# Combined Predictive Value of Extracellular Fluid/Intracellular Fluid Ratio and the Geriatric Nutritional Risk Index for Mortality in Patients Undergoing Hemodialysis

**DOI:** 10.3390/nu11112659

**Published:** 2019-11-04

**Authors:** Takahiro Yajima, Kumiko Yajima, Hiroshi Takahashi, Keigo Yasuda

**Affiliations:** 1Department of Nephrology, Matsunami General Hospital, Gifu 501-6062, Japan; 2Department of Internal Medicine, Matsunami General Hospital, Gifu 501-6062, Japan; green_tea_1324@yahoo.co.jp (K.Y.); keigo@matsunami-hsp.or.jp (K.Y.); 3Division of Medical Statistics, Fujita Health University School of Medicine, Aichi 470-1192, Japan; hirotaka@fujita-hu.ac.jp

**Keywords:** geriatric nutritional risk index, extracellular fluid, intracellular fluid, protein-energy wasting, mortality

## Abstract

The ratio of extracellular fluid (ECF) to intracellular fluid (ICF) may be associated with mortality in patients undergoing hemodialysis, possibly associated with protein-energy wasting. We therefore investigated the relationship of the ECF/ICF ratio and the geriatric nutritional risk index (GNRI) with the all-cause and cardiovascular-specific mortality in 234 patients undergoing hemodialysis. Bioimpedance analysis of the ECF and ICF was performed and the ECF/ICF ratio was independently associated with GNRI (β = −0.247, *p* < 0.0001). During a median follow-up of 2.8 years, 72 patients died, of which 29 were cardiovascular. All-cause mortality was independently associated with a lower GNRI (adjusted hazard ratio [aHR] 3.48, 95% confidence interval [CI] 2.01–6.25) and a higher ECF/ICF ratio (aHR 11.38, 95%CI 5.29–27.89). Next, we divided patients into four groups: group 1 (G1), higher GNRI and lower ECF/ICF ratio; G2, lower GNRI and lower ECF/ICF ratio; G3, higher GNRI and higher ECF/ICF ratio; and G4, lower GNRI and higher ECF/ICF ratio. Analysis of these groups revealed 10-year survival rates of 91.2%, 67.2%, 0%, and 0% in G1, G2, G3, and G4, respectively. The aHR for G4 versus G1 was 43.4 (95%CI 12.2–279.8). Adding the GNRI alone, the ECF/ICF ratio alone, or both to the established risk model improved the net reclassification improvement by 0.444, 0.793 and 0.920, respectively. Similar results were obtained for cardiovascular mortality. In conclusion, the ECF/ICF ratio was independently associated with GNRI and could predict mortality in patients undergoing hemodialysis. Combining the GNRI and ECF/ICF ratio could improve mortality predictions.

## 1. Introduction

Protein-energy wasting (PEW) is a form of malnutrition characterized by loss of body protein (muscle mass) and fuel reserves (fat mass) due to catabolic inflammation. It is especially prevalent in patients undergoing hemodialysis, in whom it is associated with an increased risk of death [1]. The malnutrition inflammation score is the well-validated tool for screening hemodialysis patients at nutritional risk, however the score requires a subjective assessment by well-trained examiners to ensure consistent results [2]. Recently, the geriatric nutritional risk index (GNRI) was reported to be a simple and objective tool that can be used to assess nutritional status in these patients [3]. Indeed, the GNRI can be applied in the assessment of PEW in hemodialysis [4], where it is reported to be a strong predictor of both all-cause and cardiovascular-specific mortality [5,6,7]. The ratio of extracellular fluid (ECF) to intracellular fluid (ICF) measured by the BIA has been introduced as a marker that simultaneously reflects both ECF overload and malnutrition in patients undergoing hemodialysis. In this setting, the ECF increases when there is an excess of that fluid, while the ICF decreases in response to reduced body cell mass, specifically skeletal muscle mass [8,9]. Both fluid overload and malnutrition have been known to be major risk factors for morbidity and mortality in hemodialysis patients [10,11,12,13]. However, ECF overload and malnutrition may interact via inflammation due to PEW, with the fluid overload itself acting as an inflammatory stimulus. Indeed, immune activation can result from the induced translocation of bowel endotoxins into the circulation when bowel edema develops [14]. This inflammatory response, in turn, causes malnutrition through increased protein catabolism and muscle wasting [15,16]. Inflammation-induced hypoalbuminemia and increased vascular permeability can contribute to the shift in extravascular fluid and cause an overload of ECF. Finally, this fluid overload may increase vessel wall stress, thereby playing an important role in the development of atherosclerosis [17]. Recently, Kim et al., [18] reported that the ECF/ICF ratio may serve as a novel indicator for all-cause mortality and new cardiovascular events in patients undergoing hemodialysis. They also suggested that this ECF/ICF ratio may reflect the malnutrition, inflammation, and atherosclerosis that constitute core elements of PEW [1].

In this study, we aimed to investigate the association between the ECF/ICF ratio and the GNRI, and to determine the combined predictive value of both for all-cause and cardiovascular-specific mortality in patients undergoing hemodialysis.

## 2. Materials and Methods

### 2.1. Study Participants

We conducted a retrospective study of patients who had undergone hemodialysis for at least 6 months and who had undergone bioimpedance analysis (BIA). The study was conducted using the records of the outpatient clinic of Matsunami General Hospital for the period between January 2008 and December 2018. All patient data were fully anonymized prior to access, and as such, the ethics committee of our hospital waived the requirement for informed consent. This study adhered to the principles of the Declaration of Helsinki, and the study protocol was approved by the ethics committee of Matsunami General Hospital (No. 435).

### 2.2. Data Collection

The following patient data were collected from medical records: age; gender; duration of hemodialysis; histories of diabetes, hypertension, and cardiovascular disease (CVD); dry weight; and height. Diabetes was defined as a history or presence of diabetes. Hypertension was defined as systolic blood pressure ≥ 140 mmHg and/or diastolic blood pressure ≥ 90 mmHg before hemodialysis session, or taking anti-hypertensive drugs. CVD was defined as heart failure, angina pectoris, myocardial infarction, and stroke. Blood samples were collected in the supine position before hemodialysis sessions, which were conducted on a Monday or a Tuesday. Chest X-ray and BIA were performed on the same day, shortly after the hemodialysis session. Body composition was assessed by a multi-frequency (2.5–350 kHz) body composition analyzer (MLT-550N, SK Medical, Japan), using the wrist–ankle method. Composition data for total body fluid (TBF), ICF, and ECF were obtained, and the ECF/ICF ratio was then calculated. The body mass index (BMI) was calculated from dry weight and height using the following formula: BMI = dry weight (kg)/height squared (m^2^). The GNRI was also calculated as follows: GNRI = (14.89 × albumin (g/dL) + [41.7 × (dry weight/ideal body weight)]. When the dry weight exceeded the ideal body weight, we set the “(dry weight/ideal body weight)” element to 1.

### 2.3. Follow-Up Study

The study endpoints were all-cause and cardiovascular mortality. Patients were primarily grouped by the median GNRI and ECF/ICF ratio. Furthermore, they were grouped into subgroups based on combinations of the median GNRI and ECF/ICF ratio: Group 1 (G1), higher GNRI and lower ECF/ICF ratio; Group 2 (G2), lower GNRI and lower ECF/ICF ratio; Group 3 (G3), higher GNRI and higher ECF/ICF ratio; and Group 4 (G4), lower GNRI and higher ECF/ICF ratio. Patients were followed up until December 2018.

### 2.4. Statistical Analysis

Normally distributed variables are expressed as means ± standard deviations, and non-normally distributed variables are expressed as medians and interquartile ranges. Differences among the four subgroups divided by the median GNRI and the ECF/ICF ratio were compared by one-way analysis of variance or the Kruskal–Wallis tests for continuous variables or by the chi-squared test for categorical variables. Pearson’s correlation coefficient was used to analyze the correlation between GNRI and ECF/ICF ratio.

Univariate regression analysis was performed to determine factors correlated with the ECF/ICF ratio. Multivariate regression analysis was performed with the factors that were significantly associated with the ECF/ICF ratio in univariate analysis. The Kaplan–Meier method was used to estimate survival, which was analyzed using the log-rank test. Hazard ratios (HRs) and 95% confidence intervals (CIs) for all-cause and cardiovascular-specific mortality were calculated by Cox proportional hazard regression analysis. The multiple regression model included all covariates significant at *p* < 0.05 in the univariate analysis, and we report adjusted HRs.

To assess whether the accuracy of predicting mortality improved after adding the GNRI and/or ECF/ICF ratio to a baseline model, we calculated the C-index, net reclassification improvement (NRI), and integrated discrimination improvement (IDI). The C-index was defined as the area under the receiver operating characteristic curve between individual predictive probabilities for mortality and the incidence of mortality, and it was compared between the baseline model with all established risk factors and the adjusted model with the GNRI and/or ECF/ICF ratio [19]. The NRI was used as a relative indicator of the number of patients for whom the predicted mortality risk improved and the IDI was used to show the average improvement in predicted mortality risk after adding the new variables to the baseline model [20].

All statistical analyses were performed using IBM SPSS Version 21 (IBM Corp., Armonk, NY, USA). A *p*-value < 0.05 was considered statistically significant.

## 3. Results

### 3.1. Baseline Characteristics

A total of 234 patients were included in the analysis, and their characteristics are summarized in Table 1. The mean age was 65.1 ± 12.6 years old and most were male (69.2%). The median duration of hemodialysis was 0.8 (0.6–5.1) years, with hypertension (92.3%), CVD (63.2%), and diabetes (45.3%) all common. The mean BMI was 22.0 ± 3.8 kg/m^2^ and the mean cardiothoracic ratio (CTR) was 49.3% ± 5.1%.

### 3.2. The GNRI and The ECF/ICF Ratio and Their Relationship

The medians for the GNRI and ECF/ICF ratio were 94.5 (89.7–98.3) and 0.556 (0.455–0.769), respectively. The ECF/ICF ratio was significantly associated with the GNRI (*r* = −0.387, *p* < 0.0001). Multivariate regression analysis revealed that the ECF/ICF ratio was independently associated with male gender (β = 0.212, *p* = 0.0001), diabetes mellitus (β = 0.171, *p* = 0.0021), the CTR (β = 0.200, *p* = 0.0010), log C-reactive protein (CRP) (β = 0.136, *p* = 0.017), and the GNRI (β = −0.247, *p* < 0.0001) (Table 2).

### 3.3. Univariate Association of The GNRI and The ECF/ICF Ratio with Mortality

During a median follow-up of 2.8 (1.2–4.9) years, 72 patients died due to CVD (29 (40.3%)), infection (23 (31.9%)), malignancy (9 (12.5%)), and other causes (11 (15.3%)). The relationship between the GNRI and the ECF/ICF ratio with all-cause and cardiovascular-specific mortality is summarized in Table 3.

In the univariate Cox proportional hazards analysis, the GNRI and the ECF/ICF ratio were significant predictors for both all-cause and cardiovascular mortality. First, we divided patients by the median GNRI into low and high groups: the 10-year all-cause survival rates were 18.7% and 67.0%, respectively; the 10-year cardiovascular survival rates were 48.7% and 83.3%, respectively (*p* < 0.0001, all). Second, we divided patients by the median ECF/ICF ratio into low and high groups: the 10-year all-cause survival rates were 0% and 84.6%, respectively; the 10-year cardiovascular survival rates were 0% and 93.4%, respectively (*p* < 0.0001, all). Third, we divided patients by both the GNRI and ECF/ICF ratios (combined group) into G1, G2, G3, and G4 groups. Patients were getting older from G1 group to G4 group (G1: 56.5 years; G2: 62.1 years; G3: 66.8 years; G4: 70.7 years). The 10-year all-cause survival rates were 91.2%, 67.2%, 0%, and 0%, respectively; the 10-year cardiovascular survival rates were 96.9%, 81.5%, 0%, and 0% in the G1, G2, G3, and G4 groups, respectively (*p* < 0.0001, all) (Figure 1).

### 3.4. Multivariate Association of The GNRI and The ECF/ICF Ratio with Mortality

We adjusted the multivariate models for age, male gender, history of CVD, creatinine, HDL-C, phosphorus, CRP, and CTR, which were significant at the *p* < 0.05 level in the univariate analyses. The adjusted HR values obtained in the subsequent analyses are shown in Table 3 adjacent those for the corresponding univariate analyses.

When comparing the lower and higher GNRI groups, the adjusted HR values for all-cause and cardiovascular mortality were significant at 3.48 and 3.19, respectively. When comparing the lower and higher ECF/ICF ratio groups, the adjusted HR values for all-cause and cardiovascular mortality were also significant at 11.38 and 11.87, respectively. Moreover, we obtained the following adjusted HR values for all-cause and cardiovascular mortality, respectively: 9.85 (95%CI 2.20–68.67, *p* = 0.0025) and 9.07 (95%CI 0.82–203.31, *p* = 0.071) for G2 vs. G1; 27.39 (95%CI 7.27–180.60, *p* < 0.0001) and 25.44 (95%CI 4.11–504.39, *p* = 0.0001) for G3 vs. G1; 43.42 (95%CI 12.22–279.83, *p* < 0.0001) and 36.56 (95%CI 6.31–719.66, *p* < 0.0001) for G4 vs. G1; 2.78 (95%CI 1.07–8.21, *p* = 0.036) and 2.80 (95%CI 0.63–19.90, *p* = 0.19) for G3 vs. G2; 4.41 (95%CI 1.88–12.26, *p* = 0.0003) and 4.03 (95%CI 1.03–27.26, *p* = 0.044) for G4 vs. G2; and 1.58 (95%CI 0.90–2.91, *p* = 0.11) and 1.44 (95%CI 0.62–3.63, *p* = 0.41) for G4 vs. G3.

### 3.5. Model Discrimination

The results of model discrimination are shown in Table 4 for the C-index, NRI, and IDI after adding the GNRI alone, the ECF/ICF ratio alone, and both combined to an established risk model. This model included age, male gender, history of CVD, creatinine, HDL-C, phosphorus, CRP, and CTR. The C-index for all-cause mortality (0.722) improved by adding the GNRI alone (to 0.755), the ECF/ICF ratio alone (0.819), and both combined (0.834) to the established risk model; the C-index for cardiovascular mortality also improved in the same order, from 0.773 to 0.786, 0.834, and 0.841, respectively. The NRI and IDI values for all-cause and cardiovascular mortality improved similarly by adding the GNRI alone, the ECF/ICF ratio alone, and both combined to the established risk model.

## 4. Discussion

We showed that the ECF/ICF ratio, as measured by BIA, was not only independently associated with the GNRI but also effectively predicted all-cause and cardiovascular mortality in patients undergoing hemodialysis. Moreover, combining the GNRI and ECF/ICF ratio appeared to improve the predictive accuracy. These results indicate that the ECF/ICF ratio may be a promising marker of both PEW and mortality, potentially making it a key assessment parameter for patients undergoing hemodialysis.

The ECF/ICF ratio has been introduced as a marker that simultaneously reflects both ECF overload and malnutrition in hemodialysis patients. Chronic fluid overload and malnutrition are well-recognized factors contributing to morbidity and mortality in this population [10,11,12,13]. As mentioned above, volume overload and malnutrition are also important causes of PEW in kidney disease [1]. Kim et al. [18] recently reported that the ECF/ICF ratio was negatively correlated with serum albumin and positively correlated with CRP, brachial-ankle pulse wave velocity, and B-type natriuretic peptide. Based on this, they suggested that the ECF/ICF ratio might be highly-related to the malnutrition-inflammation-atherosclerosis syndrome in the pathogenesis of PEW [18]. We showed that the ECF/ICF ratio was independently associated with the CTR, log CRP, and GNRI, suggesting that it is a plausible indicator of PEW.

PEW, as stated, is a state of malnutrition characterized by a loss of muscle and fat mass due to catabolic inflammation that frequently complicates chronic kidney disease and leads to increased mortality [1]. Some previous studies have shown that loss of muscle and fat stores in the presence of inflammation leads to an increased risk of CVD mortality by promoting vascular endothelial damage [21,22,23]. In a recent study, we also reported that abdominal visceral and subcutaneous fat levels measured by computed tomography were negatively associated with risks of all-cause mortality in patients undergoing hemodialysis [24]. Elsewhere, we reported that higher skeletal muscle mass and/or higher fat mass were independently associated with reduced risks of all-cause mortality in this population [25].

Our present results not only support the hypothesis proposed by Kim et al. but also suggest the value of combining the GNRI and ECF/ICF ratio when predicting all-cause and cardiovascular mortality. Indeed, a higher ECF/ICF ratio was associated with significantly increased risks for mortality in both the low GNRI (G2 vs. G4) and the high GNRI (G1 vs. G3) groups, and the predictive accuracy for mortality improved sequentially by adding the GNRI, the ECF/ICF ratio, and both variables to the baseline model. Therefore, the ECF/ICF ratio measured by BIA may be useful for predicting the risks of all-cause and cardiovascular mortality, but combining it with the GNRI could maximize its predictive value for patients undergoing hemodialysis. BIA is now widely used in the hemodialysis setting, and as such, the ECF/ICF ratio should be added to routine evaluations.

There were several limitations to our study. First, this was a retrospective single-center study of small cohort. Second, only the GNRI and ECF/ICF ratio measurements at enrollment were used for data analysis, and any changes in those values during long-term follow-up were not considered. Third, acidosis is a cause of PEW, but bicarbonate was not measured in this study. We did not also evaluate items associated with CVD risk, such as B-type natriuretic peptide, the ankle–brachial index, or mean intima-media thickness of the carotid artery. A large prospective multicenter study that uses serial measures of the GNRI and ECF/ICF ratio is needed to validate our results.

## 5. Conclusions

The ECF/ICF ratio measured by BIA is independently associated with the GNRI and is a strong predictor of all-cause and cardiovascular mortality in patients undergoing hemodialysis. In addition, combining both the GNRI and the ECF/ICF ratio could improve the ability to predict mortality outcomes. Finally, the ECF/ICF ratio also appears to serve as a valuable marker of PEW. We recommend that the ECF/ICF ratio should be added to the parameters routinely measured for patients undergoing hemodialysis.

## Figures and Tables

**Figure 1 nutrients-11-02659-f001:**
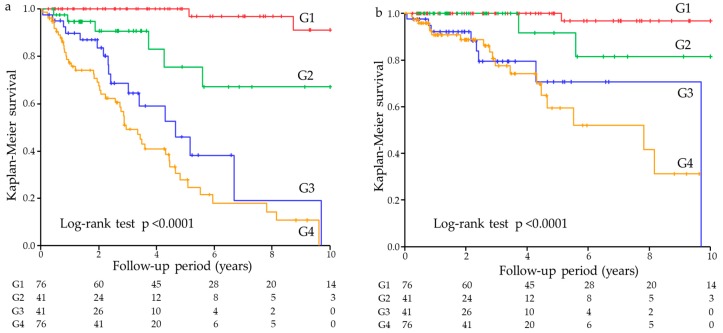
Kaplan–Meier survival curves for all-cause and cardiovascular mortality. (**a**) All-cause mortality among the four groups divided by the median GNRI and ECF/ICF ratio. (**b**) Cardiovascular mortality among the four groups divided by the median GNRI and ECF/ICF ratio. Abbreviations: GNRI, geriatric nutritional risk index; ECF, extracellular fluid; ICF, intracellular fluid; G1, higher GNRI and lower ECF/ICF; G2, lower GNRI and lower ECF/ICF; G3, higher GNRI and higher ECF/ICF; G4, lower GNRI and higher ECF/ICF.

**Table 1 nutrients-11-02659-t001:** Baseline patient characteristics.

	All Patients (*N* = 234)	G1 (*N* = 76)	G2 (*N* = 41)	G3 (*N* = 41)	G4 (*N* = 76)	*p*-Value
Age (years)	65.1 ± 12.6	56.5 ± 14.3	62.1 ± 14.3	66.8 ± 10.9	70.7 ± 10.1	<0.0001
Male (%)	69.2	65.8	48.8	75.6	80.3	0.0039
Underlying kidney disease						0.0027
Diabetic kidney disease (%)	42.7	47.3	22.0	53.7	43.4	
Chronic glomerulonephritis (%)	29.9	36.8	43.9	9.8	26.3	
Nephrosclerosis (%)	19.7	11.8	22.0	29.3	21.1	
Others (%)	7.7	4.1	12.1	7.2	9.2	
HD duration (years)	0.8 (0.6–5.1)	1.1 (0.6–5.1)	1.2 (0.5–5.7)	0.8 (0.6–3.9)	0.7 (0.6–5.7)	0.65
Alcohol (%)	20.9	17.1	19.5	29.3	21.1	0.50
Smoking (%)	25.6	27.6	23.9	24.4	25.0	0.97
Hypertension (%)	92.3	94.7	85.4	100	89.5	0.016
Diabetes (%)	45.3	47.3	26.8	58.5	46.1	0.030
History of CVD (%)	63.2	53.9	51.2	80.5	69.7	0.0061
BMI (kg/m^2^)	22.0 ± 3.8	23.3 ± 3.5	20.1 ± 4.3	23.4 ± 2.8	21.0 ± 3.7	<0.0001
BUN (mg/dL)	59.0 ± 16.8	65.9 ± 17.3	60.7 ± 13.9	57.6 ± 14.6	52.0 ± 16.0	<0.0001
Creatinine (mg/dL)	8.8 ± 3.0	10.0 ± 3.4	9.4 ± 2.8	8.0 ± 2.6	7.7 ± 2.5	<0.0001
Albumin (g/dL)	3.7 ± 0.4	3.9 ± 0.2	3.5 ± 0.3	3.8 ± 0.2	3.4 ± 0.3	<0.0001
Hemoglobin (g/dL)	10.8 ± 1.4	11.0 ± 1.3	11.0 ± 1.1	10.8 ± 1.0	10.3 ± 1.6	0.0048
T-Cho (mg/dL)	154 ± 36	158 ± 36	161 ± 32	165 ± 36	143 ± 35	0.0028
HDL-C (mg/dL)	44.5 ± 15.2	47.1 ± 16.9	46.8 ± 16.5	42.2 ± 12.4	42.0 ± 13.5	0.10
TG (mg/dL)	121 ± 77	138 ± 90	121 ± 84	136 ± 86	96 ± 40	0.0039
Uric acid (mg/dL)	7.0 ± 1.8	7.5 ± 1.7	7.5 ± 1.6	7.0 ± 1.9	6.3 ± 1.6	0.0001
Ca (mg/dL)	8.8 ± 0.8	8.9 ± 0.8	8.5 ± 0.7	9.0 ± 0.8	8.8 ± 0.9	0.047
P (mg/dL)	5.1 ± 1.3	5.5 ± 1.3	5.3 ± 1.0	4.7 ± 1.1	4.8 ± 1.5	0.0008
Glucose (mg/dL)	137 ± 58	138 ± 60	124 ± 59	158 ± 66	132 ± 48	0.050
CRP (mg/dL)	0.17 (0.07–0.48)	0.12 (0.06–0.24)	0.12 (0.03–0.67)	0.12 (0.07–0.44)	0.35 (0.13–0.92)	<0.0001
GNRI	93.5 ± 6.5	98.9 ± 3.4	89.0 ± 4.6	97.9 ± 2.2	88.2 ± 5.2	<0.0001
CTR (%)	49.3 ± 5.1	48.2 ± 4.8	48.2 ± 5.5	50.7 ± 5.7	50.2 ± 4.7	0.013
Dry weight (kg)	57.2 ± 12.1	61.5 ± 12.6	50.7 ± 12.0	60.6 ± 9.3	54.5 ± 10.7	<0.0001
Δ body weight (kg)	2.0 ± 0.9	2.4 ± 0.9	1.7 ± 0.6	1.9 ± 0.8	1.9 ± 1.0	0.0002
TBF (kg)	27.6 ± 5.5	28.4 ± 5.8	24.3 ± 5.1	28.8 ± 3.8	27.9 ± 5.5	0.0002
ICF (kg)	17.3 ± 3.8	19.7 ± 3.9	16.7 ± 3.3	16.8 ± 2.5	15.4 ± 3.2	<0.0001
ECF (kg)	10.3 ± 3.3	8.7 ± 2.4	7.5 ± 2.1	12.0 ± 2.0	12.5 ± 3.3	<0.0001
ECF/ICF ratio	0.61 ± 0.22	0.44 ± 0.08	0.45 ± 0.08	0.72 ± 0.12	0.82 ± 0.21	<0.0001

Abbreviations: HD, hemodialysis; BMI, body mass index; BUN, blood urea nitrogen; T-Cho, total cholesterol; CRP, C-reactive protein; CTR, cardiothoracic ratio; CVD, cardiovascular disease; TBF, total body fluid; ICF, intracellular fluid; ECF, extracellular fluid; GNRI, geriatric nutritional risk index; G1, higher GNRI and lower ECF/ICF ratio; G2, lower GNRI and lower ECF/ICF ratio; G3, higher GNRI and higher ECF/ICF ratio; G4, lower GNRI and higher ECF/ICF ratio.

**Table 2 nutrients-11-02659-t002:** Regression analysis of the relationships between the extracellular fluid (ECF)/intracellular fluid (ICF) ratio and baseline variables.

	Univariate		Multivariate	
Variables	β	*p*-Value	β	*p*-Value
Age	0.430	<0.0001	0.107	0.12
Male gender	0.180	0.0059	0.212	0.0001
Diabetes	0.216	0.0009	0.171	0.0021
History of CVD	0.152	0.020	0.002	0.966
CTR	0.314	<0.0001	0.200	0.0010
Creatinine	−0.399	<0.0001	−0.133	0.051
Phosphorus	−0.310	<0.0001	−0.118	0.053
Log CRP	0.329	<0.0001	0.136	0.017
GNRI	−0.387	<0.0001	−0.247	<0.0001

Abbreviations: CVD, cardiovascular disease; CTR, cardiothoracic ratio; CRP, C-reactive protein; ECF, extracellular fluid; ICF, intracellular fluid; GNRI, geriatric nutritional risk index.

**Table 3 nutrients-11-02659-t003:** Cox proportional hazards analysis of the GNRI and the ECF/ICF ratio for mortality.

Variables	Univariate		Multivariate	
	HR (95%CI)	*p*-Value	HR (95%CI)	*p*-Value
**All-cause mortality**				
GNRI (continuous)	0.88 (0.85–0.92)	<0.0001	0.90 (0.87–0.94)	<0.0001
ECF/ICF (/0.01) (continuous)	1.05 (1.04–1.06)	<0.0001	1.04 (1.03–1.05)	<0.0001
Lower GNRI	4.23 (2.54–7.37)	<0.0001	3.48 (2.01–6.25)	<0.0001
Higher ECF/ICF	15.17 (7.56–34.95)	<0.0001	11.38 (5.29–27.89)	<0.0001
Cross-classified (vs. G1)		<0.0001		<0.0001
G2	9.31 (2.14–63.66)	0.0027	9.85 (2.20–68.67)	0.0025
G3	33.99 (9.57–216.32)	<0.0001	27.39 (7.27–180.60)	<0.0001
G4	54.70 (16.53–339.15)	<0.0001	43.42 (12.22–279.83)	<0.0001
**Cardiovascular disease mortality**				
GNRI (continuous)	0.89 (0.84–0.94)	0.0001	0.89 (0.84–0.96)	0.0018
ECF/ICF (/0.01) (continuous)	1.05 (1.03–1.06)	<0.0001	1.03 (1.02–1.05)	0.0003
Lower GNRI	3.46 (1.60–8.08)	0.0014	3.19 (1.39–7.88)	0.0056
Higher ECF/ICF	18.46 (6.20–80.24)	<0.0001	11.87 (3.65–55.91)	<0.0001
Cross-classified (vs. G1)		<0.0001		<0.0001
G2	6.52 (0.62–140.54)	0.11	9.07 (0.82–203.31)	0.071
G3	36.99 (6.53–696.71)	<0.0001	25.44 (4.11–504.39)	0.0001
G4	47.26 (9.20–872.61)	<0.0001	36.56 (6.31–719.66)	<0.0001

Abbreviations: GNRI, geriatric nutritional risk index; ECF, extracellular fluid; ICF, intracellular fluid; G1, higher GNRI and lower ECF/ICF ratio; G2, lower GNRI and lower ECF/ICF ratio; G3, higher GNRI and higher ECF/ICF ratio; G4, lower GNRI and higher ECF/ICF ratio. Cross-classified means cross-classified with GNRI and ECF/ICF ratio (vs. G1). The multivariate model included all baseline variables with *p* < 0.05 in univariate analysis (age, gender, history of cardiovascular disease, creatinine, HDL-C, phosphorus, C-reactive protein, and cardiothoracic ratio).

**Table 4 nutrients-11-02659-t004:** Predictive accuracy of the GNRI and ECW/ICW ratio for mortality.

Variables	C-Index	*p*-Value	NRI	*p*-Value	IDI	*p*-Value
**All-cause mortality**						
Established risk factors+	0.722 (0.650–0.794)	Ref.	Ref.		Ref.	
GNRI	0.755 (0.687–0.824)	0.13	0.444	0.0009	0.064	0.0001
ECF/ICF	0.819 (0.761–0.876)	0.0021	0.793	<0.0001	0.142	<0.0001
GNRI and ECF/ICF	0.834 (0.778–0.890)	0.0004	0.920	<0.0001	0.170	<0.0001
**Cardiovascular disease mortality**						
Established risk factors+	0.773 (0.676–0.871)	Ref.	Ref.		Ref.	
+GNRI	0.786 (0.691–0.880)	0.52	0.403	0.021	0.024	0.045
+ECF/ICF	0.834 (0.761–0.908)	0.061	0.787	<0.0001	0.046	0.010
+GNRI and ECF/ICF	0.841 (0.773–0.909)	0.048	0.826	<0.0001	0.061	0.0024

Abbreviations: GNRI, geriatric nutritional risk index; ECF, extracellular fluid; ICF, intracellular fluid; IDI, integrated discrimination improvement; NRI, net reclassification improvement; Ref, reference. Established risk factors included all baseline variables with *p* < 0.05 in univariate cox proportional hazards analysis (age, sex, history of cardiovascular disease, creatinine, HDL-C, phosphorus, C-reactive protein, cardiothoracic ratio).

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
