# Peer review of "Combined Predictive Value of Extracellular Fluid/Intracellular Fluid Ratio and the Geriatric Nutritional Risk Index for Mortality in Patients Undergoing Hemodialysis"

_nutrients, 2019, doi:10.3390/nu11112659_

Round 1
Reviewer 1 Report
Authors analyzed ECF/ICF ratio and GNRI in HD patients. Their results are reasonable. However, there are few concerns to address, namely, the age difference between groups and the unusually high rate of hypertension (almost 100%), since BP in HD patients is closely related with volume status.
There were two major comments to authors.
1.The first one is about the age difference of the studied groups, because age difference concerned a lot if they studied the mortality. I suggest that authors might have to mention that in the text, even though the age difference was modified to lessen the impact by sophisticated statistical method.
2.The second one is about the almosy 100% rate of hypertension. High rate of hypertension in HD patients means high rate of fluid overload in the begining. To study the ECF/ICF ratio in an almost 100% fluid overloaded patients would be doubtful.
Author Response
Response to Reviewer 1
Thank you very much for your very constructive comments.
1.The first one is about the age difference of the studied groups, because age difference concerned a lot if they studied the mortality. I suggest that authors might have to mention that in the text, even though the age difference was modified to lessen the impact by sophisticated statistical method.
Thank you very much for the comment. The comment is appropriate.
Kobayashi et al. (Nephrol. Dial. Transplant 2010,25,3361-3365) reported that age and GNRI was negatively correlated (r = −0.436, P < 0.0001), and that the age was significantly older in lower GNRI group than in higher GNRI group (70 year vs 59 year, p <0.0001). On the other hand, Kim et al. (PLOS ONE 2017,12,e0170272) also reported that the age of higher ECF/ICF ratio group was significantly older than that of lower ECF/ICF ratio group (57.3 years vs 49.3 years, p = 0.005). In the present study, as you mentioned above, the age increased in order when the risks of all-cause and cardiovascular mortality increased. Thus, we performed multivariate analysis including age to adjust the difference. In this context, we think it is not surprising but natural that the age is older in higher risk group than in lower risk group. Of course, because the age difference is important factor for clinical prognosis, we mentioned this in the revised manuscript.
2.The second one is about the almosy 100% rate of hypertension. High rate of hypertension in HD patients means high rate of fluid overload in the begining. To study the ECF/ICF ratio in an almost 100% fluid overloaded patients would be doubtful.
Thank you very much for the comment.
However, we think that hypertension is not always associated with fluid overload. Many hemodialysis patients are under the risk for fluid overload. At the initiation of hemodialysis, many patients complicated with hypertension and fluid overload. However, they were generally set at dry weight, and then anti-hypertensive medication may be prescribed if there is hypertension even after dry weight is appropriately decided. Moreover, in the present study, bio-impedance analysis obtaining ECF/ICF ratio was performed after the hemodialysis session (after removing water until dry weight). Therefore, we basically evaluated ECF/ICF ratio in the situation of not being apparent volume overload.
Reviewer 2 Report
1. Does the study period end up in 2017 (section 2.1) or 2018 (section 2.3)?
2. The physiological meanings of both GNRI and ECF/ICF should be stated better in the introduction instead of the discussion section. The Pathophysiology of mortality in patients under going hemodialysis should also be discussed in the introduction section. Review of the published studies could be more extensive.
3. As most of the subjects have hypertension, were they under anti-hypertensive or other medications? Did the blood pressure or any medication affect the mortality risk?
4. How did you choose the cut points to make the four groups? just using the median?
5. Although GNRI and ECF/ICF are significantly associated, we still find in group three 41 subjects (quite a lot) are high in GNRI but low in ECF/ICF. The more crucial factor may be age, because we also find from group one to group four, the patients are getting older. Why did you choose GNRI instead of age, or even CRP, Cre, etc for the prediction model?
Author Response
Response to Reviewer 2
Thank you very much for your very constructive comments.
Does the study period end up in 2017 (section 2.1) or 2018 (section 2.3)?
Thank you very much for the comment.
The study period end up in 2018. It was a simple mistake. We are sorry. We revised our manuscript.
The physiological meanings of both GNRI and ECF/ICF should be stated better in the introduction instead of the discussion section. The Pathophysiology of mortality in patients under going hemodialysis should also be discussed in the introduction section. Review of the published studies could be more extensive.
Thank you very much for the comment. According to your advice, we revised our manuscript.
As most of the subjects have hypertension, were they under anti-hypertensive or other medications? Did the blood pressure or any medication affect the mortality risk?
Thank you very much for the comment.
In the present study, 95.7% of hemodialysis patients had a history of hypertension. At the study enrollment, 202 patients (86.3%) were taking anti-hypertensive medications [ARB: 109 (46.6%); ACE-I: 14 (6.0%); CCB: 164 (70.1%); diuretics: 100 (42.7%); β blocker: 60 (25.6%)]. Even if hypertension was defined as systolic blood pressure >140 mmHg and/or diastolic blood pressure >90 mmHg before dialysis session, or taking anti-hypertensive medications, 92.3% had hypertension. However, newly defined hypertension, the use of anti-hypertensive medications, and any class of anti-hypertensive medications were not associated with mortality [newly defined hypertension: Hazard ratio (HR) 0.81 95% confidence interval (CI) 0.36-2.31; the use of anti-hypertensive medications: HR 0.64 (95%CI 0.34-1.32); ARB: HR 0.90 (95%CI 0.56-1.42); ACE-I: HR 0.90 (95%CI 0.40-2.58); CCB: HR 0.81 (95%CI 0.50-1.34); diuretics: HR 0.89 (95%CI 0.54-1.44); β blocker: 0.99 (95%CI 0.56-1.67].
How did you choose the cut points to make the four groups? just using the median?
Thank you very much for the comment.
The cut-off points for divided into four groups were just using the medians of GNRI and ECF/ICF ratio.
Although GNRI and ECF/ICF are significantly associated, we still find in group three 41 subjects (quite a lot) are high in GNRI but low in ECF/ICF. The more crucial factor may be age, because we also find from group one to group four, the patients are getting older. Why did you choose GNRI instead of age, or even CRP, Cre, etc for the prediction model?
Thank you very much for the comment. The comment is appropriate.
Kobayashi et al. (Nephrol. Dial. Transplant 2010,25,3361-3365) reported that age and GNRI was negatively correlated (r = −0.436, P < 0.0001), and that the age was significantly older in lower GNRI group than in higher GNRI group (70 year vs 59 year, p <0.0001). On the other hand, Kim et al. (PLOS ONE 2017,12,e0170272) also reported that the age of higher ECF/ICF ratio group was significantly older than that of lower ECF/ICF ratio group (57.3 years vs 49.3 years, p = 0.005). In the present study, as you mentioned above, the age increased in order when the risks of all-cause and cardiovascular mortality increased. Thus, we performed multivariate analysis including age to adjust the difference. In this context, we think it is not surprising but natural that the age is older in higher risk group than in lower risk group. Of course, because the age difference is important factor for clinical prognosis, we mentioned this in the revised manuscript. Moreover, in this study, we focused the association between the ECF/ICF ratio and the GNRI as a marker of the PEW, and to determine the combined predictive value of both for all-cause and cardiovascular-specific mortality in patients undergoing hemodialysis. Thus, age, CRP or creatinine were set as covariates as established risk factors already, and were entered into multivariate models, of course. We believe you may understand this.
Reviewer 3 Report
Basically, this retrospective analysis of BIA data from HD patients using parameters that are easy to determine is a wonderful idea. The study is clearly structured and reads very understandably, also the message to use GNRI and ECF/ICF ratio is obvious and comprehensible. Therefore, this article is a valuable addition to other articles that have dealt with this topic. The Kaplan-Meier survival curves are impressive for themselves.
Criticisms:
In my opinion, it is not surprising that ~ 14 years older patients (G4: 70.7 years versus G1: 56.5 years) have a significantly higher risk of mortality, regardless of the ECF / ICF ratio and the GNRI, especially as comorbidities like cardiovascular disease are substantially higher in this group. Therefore Table 1 (Baseline) basically shows the major aging events with the onset of substance loss. Thus, not only albumin but also uric acid and urea and lipids are correspondingly reduced. Albumin itself is an integral part of the formula for GNRI, and must therefore be predictive. In this regard, it is necessary to look at the collectives once again by age group with the indices. I understand too little about math, but in multivariate analysis, I would have expected that age to be clear
In addition, the underlying nephrological diseases should be supplemented
Bicarbonate should be added and investigated as metabolic markers
Author Response
Response to Reviewer 3
Thank you very much for your very constructive comments.
In my opinion, it is not surprising that ~ 14 years older patients (G4: 70.7 years versus G1: 56.5 years) have a significantly higher risk of mortality, regardless of the ECF / ICF ratio and the GNRI, especially as comorbidities like cardiovascular disease are substantially higher in this group. Therefore Table 1 (Baseline) basically shows the major aging events with the onset of substance loss. Thus, not only albumin but also uric acid and urea and lipids are correspondingly reduced. Albumin itself is an integral part of the formula for GNRI, and must therefore be predictive. In this regard, it is necessary to look at the collectives once again by age group with the indices. I understand too little about math, but in multivariate analysis, I would have expected that age to be clear
Thank you very much for the comment. The comment is appropriate.
Kobayashi et al. (Nephrol. Dial. Transplant 2010,25,3361-3365) reported that age and GNRI was negatively correlated (r = −0.436, P < 0.0001), and that the age was significantly older in lower GNRI group than in higher GNRI group (70 year vs 59 year, p <0.0001). On the other hand, Kim et al. (PLOS ONE 2017,12,e0170272) also reported that the age of higher ECF/ICF ratio group was significantly older than that of lower ECF/ICF ratio group (57.3 years vs 49.3 years, p = 0.005). In the present study, as you mentioned above, the age increased in order when the risks of all-cause and cardiovascular mortality increased. Thus, we performed multivariate analysis including age to adjust the difference. In this context, we think it is not surprising but natural that the age is older in higher risk group than in lower risk group. For these reasons, we think, for example, a statistically age-matching method is actually inadequate in this study.
In addition, the underlying nephrological diseases should be supplemented
Thank you very much for the comment.
We added the underlying kidney diseases to the table 1.
Bicarbonate should be added and investigated as metabolic markers
Thank you very much for the comment.
Acidosis is one of the important cause of the protein-energy wasting (Kidney int. 2008,73,391-398). Unfortunately, bicarbonate was not measured in this study, because the present study was retrospectively performed. However, this comment is important. We add this in the limitation section of the revised manuscript.
Round 2
Reviewer 1 Report
Authors have revised their text and responded to the comments.
Author Response
Thank you very much.
Reviewer 2 Report
1. A more extensive review of previous publications will be better.
2. The main purpose to develop a new parameter for a prognostic prediction is that it makes better prediction. It would be better if you can show that combining GNRI and ECF/ICF ratio can predict better then either age, CRP or creatinine.
3. Why did you correct the data of the percentage of hypertension in each group?
Author Response
Response to Reviewer 2
Thank you very much for your constructive comments.
A more extensive review of previous publications will be better.
Thank you very much for the comment.
We included previous publications enough to understand our paper.
The main purpose to develop a new parameter for a prognostic prediction is that it makes better prediction. It would be better if you can show that combining GNRI and ECF/ICF ratio can predict better then either age, CRP or creatinine.
Thank you very much for the comment.
We have already shown that the C-index for all-cause and cardiovascular mortality significantly improved by adding the combined GNRI and ECF/ICF ratio to the established risk model including age, CRP, and creatinine. Similar results were obtained for NRI and IDI. You can check this in the result section of model discrimination and in Table 4.
Why did you correct the data of the percentage of hypertension in each group?
Thank you very much for the comment.
As you mentioned, the percentage of hypertension was high in our study. So, we reviewed blood pressure before hemodialysis session and anti-hypertensive medications in all patients. And, we found that the definition of hypertension was ambiguous. Therefore, we reanalyzed data after hypertension was defined as systolic blood pressure ≥ 140 mmHg and/or diastolic blood pressure ≥ 90 mmHg before hemodialysis session, or taking anti-hypertensive drugs.